# Distribution of Bioelectrical Impedance Vector Analysis and Phase Angle in Korean Elderly and Sarcopenia

**DOI:** 10.3390/s23167090

**Published:** 2023-08-10

**Authors:** Fang-Lin Jiang, Saizhao Tang, Seon-Ho Eom, Jae-Young Lee, Ji Heon Chae, Chul-Hyun Kim

**Affiliations:** 1Department of Wushu and Traditional National Sports, College of Physical Education, Hunan Normal University, Changsha 410012, China; jfl0108681@gmail.com; 2Department of Sports Medicine, Soonchunhyang University, Asan 31538, Republic of Korea; tangsaizhao@gmail.com (S.T.); ush1995@sch.ac.kr (S.-H.E.); lhsljy1@sch.ac.kr (J.-Y.L.); kb062344@gmail.com (J.H.C.)

**Keywords:** bioelectrical impedance vector analysis, phase angle, aging, diagnosis, noninvasive

## Abstract

Purpose: This study aimed to verify whether bioelectrical impedance vector analysis (BIVA) can support the clinical evaluation of sarcopenia in elderly individuals and evaluate the relationships between phase angle (PhA), physical performance, and muscle mass. Methods: The sample comprised 134 free-living elderly individuals of both sexes aged 69–91 years. Anthropometric parameters, grip strength, dual-energy X-ray absorptiometry findings, bioimpedance analysis results, and physical performance were also measured. The impedance vector distributions were evaluated in elderly individuals using BIVA. Results: BIVA revealed significant differences between the sarcopenia and non-sarcopenia groups (both sexes). The sarcopenia group had a significantly lower PhA than the non-sarcopenia group in both sexes (*p* < 0.05). PhA was significantly correlated with age, appendicular skeletal muscle (ASM), handgrip strength (HGS), and muscle quality in both sexes and significantly correlated with ASM/Height^2^ and physical performance in males. Conclusion: BIVA can be used as a field assessment method in elderly Koreans with sarcopenia. PhA is a good indicator of muscle strength, muscle quality, and physical performance in males. These methods can help diagnose sarcopenia in elderly individuals with reduced mobility.

## 1. Introduction

According to the World Population Prospects [1], in 2017, approximately 962 million people were over 60 years of age—this number is projected to increase to 1.4 billion by 2030 and 2.1 billion by 2050. In addition, the population aged > 80 years is projected to increase from 143 million to 426 million by 2050. According to South Korea’s “Population Trends and Projections of the World and Korea”, the global population aged older than 65 years is expected to increase from 8.2% in 2015 to 17.6% in 2060, while in South Korea, it is expected to increase from 13.1% in 2015 to 40.1% by 2060 [2]. The proportion of elderly with limited mobility, frailty, and adverse outcomes (e.g., falls, fractures, and loss of autonomy) is expected to increase correspondingly [3].

Muscle health changes with age. Age-related loss of skeletal muscle mass and function—sarcopenia—is associated with frailty and increased risk (increasing by approximately 1%/year), reaching approximately 50% by the age of 80 years and becoming a morbidity (chronic disease). Approximately 5–13% of people aged 60–70 years and 11–50% of people aged over 80 years are affected by sarcopenia [4,5,6]. Among adults aged over 60 years in the United States, the direct healthcare costs due to sarcopenia in 2000 were estimated at USD 18.5 billion [7]. Janssen et al. (2004) [7] showed that a 10% decrease in the incidence of sarcopenia reduced healthcare costs by USD 1.1 billion annually. Therefore, the prevention and treatment of this condition are urgent public health issues worldwide.

Sarcopenia was diagnosed according to the guidelines proposed by the European Working Group on Sarcopenia in the Elderly (EWGSOP) in 2010, which were updated in 2018 (EWGSOP2). The diagnosis of sarcopenia is based on three criteria: muscle strength, muscle quantity or quality, and physical performance [8].

Although there are several methods for evaluating muscle mass, each has limitations. Computed tomography (CT) and magnetic resonance imaging (MRI) are reference methods for measuring muscle mass [9,10]. Whole-body dual-energy X-ray absorptiometry (DXA) is widely used in the clinical setting. This method provides accurate and efficient measurements of whole-body and organ/tissue composition but is very expensive and cannot be used in most clinical or field settings [11]. Sarcopenia can be assessed using physical performance tests; however, this is challenging for elderly people with limited mobility. Therefore, clinical methods that can easily and quickly assess muscle mass and physical function are required.

Bioimpedance analysis (BIA) is a non-invasive, easy-to-use, inexpensive, and portable method that has been used to assess body composition for decades. Using the impedance value obtained by BIA, participant’s height, weight, sex, etc., in a regression equation, lean body mass (LBM) and total body water (TBW) can be calculated [12].

BIA does not directly measure the TBW, body fat, or LBM. Instead, it measures electrical resistance (*R*) and reactance (*Xc*). Impedance (*Z*) is the sum of these two variables. *R* is negatively proportional to the electrolytes in the body fluid and living tissues. *Xc* is proportional to the capacitance of the cell membrane and depends on the function, composition, and structural integrity of the cell [13,14]. Traditionally, the BIA value is calculated as the square of the participant’s height divided by the impedance. It is used to quantitatively infer body composition using a regression equation that includes empirical variables such as weight, sex, and height. In addition, during the development of these predictive regression equations, several variables must be considered to ensure measurement accuracy. Specifically, body shape (approximated as five connected cylinders) and hydration level (the observed ratio of TBW to free fat mass (FFM): 73%) must be considered; the regression equations are based on the assumption that each body part has the same impedance coefficient (body impedance ratio) [15,16].

Although the estimation of body composition makes use of empirical variables (sex, race, age, etc.) in the BIA regression formula, by definition, this formula is only suitable for use with the specific target groups for which it was developed, which limits its application and hinders generalization. Thus, BIA can only be used correctly after verifying its validity for a given instance [17,18].

Unlike the general public, children and elderly individuals with poor eating habits, very poor nutritional status, or severe diseases or conditions (e.g., cancer, heart disease, thyroid disease, anorexia, or sarcopenia) have different body compositions that impair BIA calculations [19,20]. Due to errors in the bioelectrical variables generated during the process of estimating body composition, the values computed by the BIA regression equation are incorrect. To overcome this limitation, bioelectric variables can be used directly to evaluate body composition and changes in physiological status.

Bioelectric variables at 50 kHz include resistance and induction resistance, the relationships between them indicate the physiological state of the human body. The variables directly measured by 50 kHz phase-sensitive BIA are resistance and sensory reactance. The phase angle (PhA) obtained by dividing *Xc* (obtained from BIA) by *R* can indicate the stable and pathological states of the cells. For example, as PhA decreases, *Xc* decreases relative to *R*. In contrast, an increase in PhA indicates an increase in *Xc* relative to *R*, which may reflect an improvement in biological cellular functions [21,22].

Recent studies have re-evaluated the use of PhA to assess cell health, particularly for detecting defects in cell membranes and cell function. Although accumulating evidence suggest that PhA can serve as an indicator of nutritional status [23], disease prognosis [24], and mortality risk [25], whether it indicates muscle function or quality remains uncertain.

Bioimpedance vector analysis (BIVA) is a BIA-derived method that utilizes raw impedance parameters to evaluate physiological states, facilitating the assessment of somatic cell mass and hydration without referring to predictive equations [26]. This method has been used to assess many pathological conditions [27] and is highly sensitive to changes in handgrip strength.

These results provide new evidence for the validity of 50 kHz phase-sensitive bioelectrical measurements as biomarkers of hydration status, catabolic states, and nutritional status. To enable the development of clinical research and practice in sarcopenia, it is essential to establish the distribution of BIVA and phase angle for a population and disease status. However, validation studies that use BIVA and phase angle to identify individuals with sarcopenia in the Korean population are currently lacking.

This study aimed to identify the distribution of BIVA and comparison of the phase angle between Korean elderly and patients with sarcopenia in an attempt to identify whether BIVA and phase angle at 50 kHz could be innovative, useful biomarkers in identifying sarcopenia in the Korean population.

## 2. Materials and Methods

### 2.1. Participants

Participants were recruited through local newspapers and social media advertisements. Participants who were registered at public health centers in the boroughs and who volunteered to participate were also included. The study was approved by the Ethics Committees of the Korean National Sport University (No. 1263-201903-HR-010-02) and Soonchunhyung University (1040875-2020307-BC-083) and was performed in accordance with the Declaration of Helsinki.

The inclusion criteria were as follows: age of 65 years or more with independence and autonomy in daily life. Participants were evaluated as subjectively healthy based on declarations of good health and no limitations in daily activities. Of the 220 participants registered for the test, 74 were excluded from the study. The exclusion criteria were as follows: those who did not undergo any tests, limb amputation, metal prosthetic devices, or electrically based implants.

### 2.2. Anthropometric Measurements

During all measurements, the subjects were instructed to wear light clothing and remove all metallic objects, jewelry, and shoes. Measurements were obtained on the same day after fasting for at least 4 h. Body weight was measured in units of 0.5 kg (CAS DB-1, 106, CAS, Seoul, Republic of Korea), and height was measured in 1 mm units (SECA 274, 107, seca, Hamburg, Germany).

### 2.3. Measurement of the Bioelectrical Impedance Parameters

Quantum Desktop RJL-101 (RJL Systems, Clinton Twp, MI, USA), a 4-point electrode real-time BIA device, was used to assess impedance parameters. Participants were asked to fast overnight, refrain from strenuous physical activity for 12 h, and avoid drinking alcohol for 24 h prior to measurement. The last meal was completed at least 4 h before measurement, and the bladder was emptied within 30 min of measurement. Bioelectric variables were measured by skilled operators according to international standards [28]. Each evaluation was performed using the same tester. The examiner’s intraclass correlation coefficient (ICC) for repeatability was determined using a single test. The intraclass correlation coefficients of the examiner for repeatability were 0.961 and 0.955, respectively, as previously reported [28].

The adhesive gel electrodes were placed at defined anatomical sites on the anterior dorsal surfaces of the right hand, wrist, ankle, and feet. For the hand and wrist, the proximal edge of the detection (voltage) electrode was attached to form an imaginary line that bisected the styloid process of the ulna, and the proximal edge of the source (current) electrode bisected the metacarpophalangeal joint of the middle finger. For the ankle and foot, the proximal edge of the ankle-detecting electrode was attached to form an imaginary line bisecting the medial malleolus, and the distal edge of the foot-source electrode was placed to form an imaginary line passing through the metatarsophalangeal joints of the second and third toes [28].

### 2.4. Definition of Sarcopenia

Sarcopenia was defined as low muscle strength, low muscle mass, and/or low physical performance according to the EWGSOP2 guidelines [29,30,31]. The appendicular skeletal muscle (ASM) was assessed using DXA. All centers used the same DXA model (Lunar DPX-L model, version 3.4) for body composition measurements. The measurements were performed in the medium scan mode, with the subject lying in a supine position. The scanning time was approximately 20 min. In the Korean population, low muscle mass was defined as DXA < 7.0 kg/m^2^ in males and <5.9 kg/m^2^ in females [30]. Grip strength was measured using a grip dynamometer. Muscle strength was assessed through handgrip testing using a grip dynamometer (Grip-D; Takei, Niigata, Japan). Strength was recorded in kilograms (kg). The left and right hands were tested three times, and the highest values were used [29]. Low muscle strength was defined as a handgrip strength of <28 kg for males and <18 kg for females in the Korean population. Physical performance was assessed by asking the participants to walk 6 m at a comfortable pace. The test began when the first foot crossed the starting line and stopped when the second foot crossed the finish line [29]. The 6 m walk was performed two times, and the highest value was used. The criterion for low physical performance was a 6 m walk speed of <1.0 m/s [31].

### 2.5. Data Processing and Statistical Analysis

Descriptive, univariate, and bivariate statistical analyses (mean, standard deviation (SD), and Pearson’s correlation coefficient) were performed using SPSS version 23.0 (IBM Corporation, Armonk, NY, USA). BIVA includes *R* and *Xc* values (*Z* vector). Raw values were normalized to subject height (*R*/*H* and *Xc*/*H*) and presented in an *R-Xc* plot constructed using BIVA software (version 2002, Piccoli A, Pastori G, available from apiccoli@unipd.it (accessed on 14 July 2006)). The normal intervals of the reference population were expressed in percentiles (50th, 75th, and 95th) of a Gaussian bivariate probabilistic graph [31,32]. Finally, the parameters required to generate a dimensionless *R-Xc* score graph for the Korean population were calculated using the formula described in a previous study [33]. Statistical significance was set at *p* < 0.05. All tests were 2-tailed.

## 3. Results

A total of 146 participants—males and 71 females aged 69–91 (Table 1)—were included in this study. The bivariate 95% confidence ellipses of the mean vectors were calculated using the descriptive statistics reported in Table 1 (i.e., the mean and SD of the vector components *R*/*H* and *Xc*/*H* and the correlation coefficient of the two components), as defined in BIVA. Males had significantly greater height, weight, ASM, ASMI, muscle quality, and PhA than females (*p* < 0.001), while females had significantly higher *R*, *Xc*, *R*/*H*, and *Xc*/*H* values than males (*p* < 0.05).

As shown in Table 2, in the non-sarcopenia group, males had lower impedance (444.9 Ω in males and 536.7 Ω in females) and higher PhA than females (6.0° in males and 5.3° in females). In sarcopenia, males had lower impedance (513.7 in males and 655.0 in females) and higher PhA than females (5.2 in males and 4.7 in females). Men in the sarcopenia group had higher impedance (513.7 Ω vs. 444.9 Ω) and lower PhA (5.2 ± 0.5° vs. 6.0 ± 0.6°) than those in the non-sarcopenia group. The same trend was observed in women. The PhA of healthy elderly individuals was higher than that of sarcopenic elderly individuals in both males (6.0 ± 0.6° vs. 5.2 ± 0.5°) and females (5.30 ± 0.8° vs. 4.7 ± 0.8°).

The bivariate 95% confidence ellipses of mean vectors were calculated using the descriptive statistical measures reported in Table 3 (i.e., the mean and SD of vector components *R/H* and *Xc*/*H* and the correlation coefficient between the components) as defined through BIVA methods. The height-adjusted *R* significantly differed between the sarcopenia and non-sarcopenia groups, with the *R*/*H* being significantly higher in the sarcopenia than in the non-sarcopenia group. However, no significant differences in height-adjusted *Xc* were observed between the groups in either men or women. In addition, a significant difference in PhA was observed between the sarcopenia and non-sarcopenia groups, with the sarcopenia group having a lower PhA in males (*p* < 0.01) and females (*p* < 0.05).

Figure 1 shows the results of connecting the two centers of adjacent 95% confidence ellipses for males and females. The spatial distribution of the mean vectors in the *R*-*Xc* plane followed the same pattern for both sexes. Both men and women in the sarcopenia group had longer impedance vectors.

For each sex, 50%, 75%, and 95% tolerance ellipses (i.e., the intervals in which a vector from an individual subject falls with probabilities of 50%, 75%, and 95%, respectively) were calculated using the data reported in Table 3 and are depicted as *R*-*Xc* graphs (Figure 2). The size of the tolerance ellipses was determined by the variation in both vector components (larger ellipses were produced by groups with greater SD), and the elliptical shape was defined by the correlation coefficients between the vector components.

With BIVA, intersubject variability in the impedance vector is represented by a bivariate normal distribution, that is, a graph with elliptical probability regions (50%, 75%, and 95% tolerance ellipses) on the *R*-*Xc* plane, normalized by patient height (*R*/*H* and *Xc*/*H*, in ohm/m). The vector position on the *R*-*Xc* graph is interpreted and ranked in the following two directions: (1) vector displacements parallel to the major axis of the tolerance ellipses indicate progressive changes in soft tissue hydration, and (2) vectors lying on the left side and above the major axis or on the right side and below the major axis of the tolerance ellipses indicate more or less soft tissue mass, respectively. *R*: Resistance; *Xc*: reactance; and *H*: height [26].

Table 4 shows the correlation coefficients of PhA with age and ASM in males and females, respectively. PhA correlated with age, ASM, HGS, and muscle quality in both males and females; PhA correlated with ASM/H^2^ and physical performance in men. The correlation coefficients for males were as follows: age (*r* = −0.275, *p* < 0.05), ASM (*r* = 0.229, *p* < 0.05), ASM/Ht^2^ (*r* = 0.352, *p* < 0.01), HGS (*r* = 0.350, *p* < 0.01), physical performance (*r* = 0.493, *p* < 0.001), and muscle quality (*r* = 0.272, *p* < 0.05). In females, they were age (*r* = −0.389, *p* < 0.01), ASM (*r* = 0.271, *p* < 0.05), HGS (*r* = 0.640, *p* < 0.001), and muscle quality (*r* = 0.524, *p* < 0.001).

PhA was a significant indicator of muscle strength in both males (β = 2.6; *p* < 0.01) and females (β = 3.4; *p* < 0.001) as well as muscle quality in both males (β = 0.07; *p* < 0.05) and females (β = 0.17; *p* < 0.001). In males, PhA was a significant indicator of physical performance (β = 0.3; *p* < 0.001).

Table 5 shows the linear regression analysis between the phase angle and the dependent variables including muscle strength, muscle quality and physical performance in males and females, respectively. PhA had determinant correlations with muscle strength, muscle quality, and physical performance, respectively (male: *R^2^* = 0.12, *R^2^* = 0.07, *R^2^* = 0.24; female: *R^2^* = 0.41, *R^2^* = 0.27, *R^2^* = 0.005)

The results of the multiple linear regression analyses are presented in Table 6. In males, Model 1 included physical performance and PhA, with the latter being the dependent variable. Physical performance was positively associated with PhA. Model 2 included physical performance, ASM/H^2^, and PhA. Model 3 included physical performance, the ASM/H^2^ ratio, muscle quality, and PhA score. All models (Models 1–3) had weak *R*^2^ values of 0.243, 0.303, and 0.347, respectively. In females, Model 1 included muscle quality and PhA, whereas Model 2 included muscle quality, ASM, and PhA. Models 1 and 2 exhibited weak *R*^2^ values of 0.274 and 0.439, respectively.

Figure 3 shows that the sarcopenia group had a significantly lower PhA than the non-sarcopenia group in both males (*p* < 0.01) and females (*p* < 0.05). The PhA of males was higher than that of females.

## 4. Discussion

This study used the AWGS criteria to diagnose sarcopenia in elderly Koreans, established BIVA reference values in 146 healthy elderly Koreans aged 69–91 years, and determined whether BIVA can be used to diagnose sarcopenia in individuals. In addition, the correlations between PhA and physical performance, muscle quality, and other factors were determined.

Individuals with sarcopenia exhibited different bioelectrical properties than healthy older adults, as found with BIVA. In particular, individuals with sarcopenia had lower PhAs and longer impedance vectors than individuals without sarcopenia. This was consistent with the theoretical expectations of BIVA, as the mean vector of sarcopenia was in this region, with *R-Xc* corresponding to lean individuals [26]. In general, a low PhA is associated with a low somatic mass [26] and a high extracellular water/intracellular water ratio [34]. Additionally, Castillo-Martínez et al. [32] found low PhA, low *Xc*/*H*, and high *R*/*H* values in individuals with cachexia.

Norman et al. observed similar impedance vector shifts in patients with lower grip strength. The authors interpreted this displacement as low cellular and muscle function and suggested that BIVA could be assessed instead of grip strength in certain patients [33]. The same impedance vector shift was observed in a study on elderly Italian individuals [34].

The length of the BIVA vector represents the hydration of soft tissue [26,35,36]. Within the reference range, the sex-specific 75% tolerance ellipse indicates normal hydration, short vectors below this limit indicate overhydration, and long vectors above the 75% tolerance ellipse indicate underhydration [26]. This explains why most individual impedance patterns in patients with sarcopenia appear outside the 75% tolerance ellipse.

We found that females had lower PhAs than males. PhA was negatively correlated with age and positively correlated with ASM, ASMI (in males), HGS, physical performance (in males), and muscle quality. Furthermore, in older males, PhA was independently associated with ASMI, muscle quality (HGS/ASM), and physical function. In older females, PhA was independently associated with muscle mass and ASM. The PhA was significantly lower in the sarcopenia than in the non-sarcopenia group.

Females have a lower PhA than males, and PhA decreases with age [15,32,33,36,37]. PhA has been positively correlated with muscle mass [38], HGS [39], lower-extremity muscle function [40], and physical performance [41] (assessed by an individual’s walking speed and ability to rise from a seated position).

Furthermore, the PhA can be used to predict physical performance [39]. Although these results are consistent with previous reports, we found that the strongest association between physical function and PhA was observed in older men, whereas the strongest association between HGS and PhA was observed in older women. A previous study demonstrated a weak-to-moderately positive association between PhA and muscle quality (HGS/USM) in both men and women [40]. The present study found that PhA and muscle quality (HGS/ASM) were moderately positively correlated in both males and females. Multiple linear regression analysis showed that PhA was significantly associated with physical function, ASMI, and muscle mass in older men. In older females, the PhA was significantly associated with muscle mass and ASM.

Other studies have shown that PhA is associated with high cell counts, good cell function, and membrane integrity [41,42,43,44]. Loss of muscle mass results in a decrease in reactance, which causes a decrease in intracellular water, leading to an increase in impedance, which in turn causes a decrease in PhA. Increases in intramuscular fat and fibrous tissue lead to decreased muscle quality [45]. Therefore, muscle quality and quantity are independently associated with PhA. Considering that PhA measurements can determine muscle mass and function, they may be an alternative to expensive methods of diagnosing sarcopenia.

Our findings can help identify older adults showing early signs of adverse health conditions such as frailty, muscle loss, prolonged hospital stays, and reduced mobility, facilitating the early prevention of sarcopenia to avoid later health issues. In the future, it will be necessary to expand the sample size to further study the relationship between PhA and physical function to verify whether PhA can be used as an explanatory factor for physical function in elderly Korean women.

This study had several limitations. First, the sample size was relatively small. Because impedance values are affected by age, specific age-stratified population reference values (e.g., age groups 69–79 and 80–89) should be considered when establishing BIVA reference values. Second, the participants in this study were elderly individuals who were able to live independently and were relatively healthy. Therefore, the prevalence of sarcopenia in older adults in this study was lower than that previously reported. Third, the scarcity of patients with sarcopenia did not allow the establishment of a cutoff point for PhA. In the future, the study cohort should be expanded to obtain additional bioelectrical impedance data for elderly Koreans with sarcopenia. This would allow for the determination of diagnostic reference values for impedance parameters in the sarcopenia population. Finally, these reference data are applicable only to specific devices and populations. Caution should be exercised when applying this method to other populations and devices.

## 5. Conclusions

Sarcopenia imposes high costs on health services, severely lowers the quality of life of older adults, and is currently a public health problem. It is important to identify individuals at risk of sarcopenia quickly and easily for timely prevention and treatment measures. This study aimed to determine whether BIVA and PhA were suitable tools for the assessment and diagnosis of sarcopenia in the Korean population.

This study established BIVA reference data for Korean elderly using 50 kHz phase-sensitive equipment, as well as for detecting changes in muscle mass in Korean elderly patients with sarcopenia, providing evidence that BIVA can be used as a field assessment method for Korean elderly individuals with sarcopenia. It also established PhA as an indicator of muscle strength and muscle quality, and also physical performance in men. This is helpful in diagnosing sarcopenia in elderly individuals with reduced mobility. Compared with the diagnostic criteria for sarcopenia, BIA-derived BIVA and PhA are simpler, more convenient, and more economical for predicting and diagnosing sarcopenia.

## Figures and Tables

**Figure 1 sensors-23-07090-f001:**
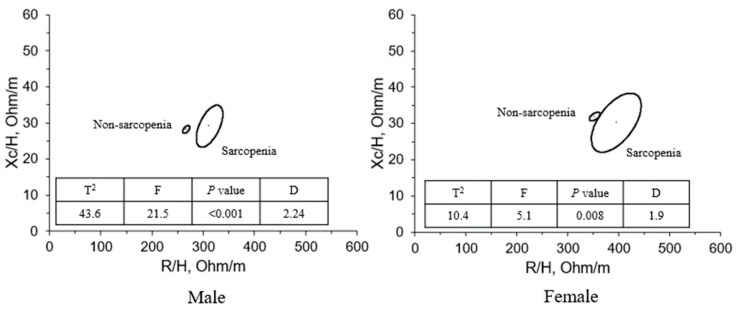
Mean impedance vectors and 95% confidence ellipses in the sarcopenia and non-sarcopenia groups. *R*, resistance; *Xc*, reactance; T^2^, Hotelling’s 2 F, F statistic; D, Mahalanobis distance.

**Figure 2 sensors-23-07090-f002:**
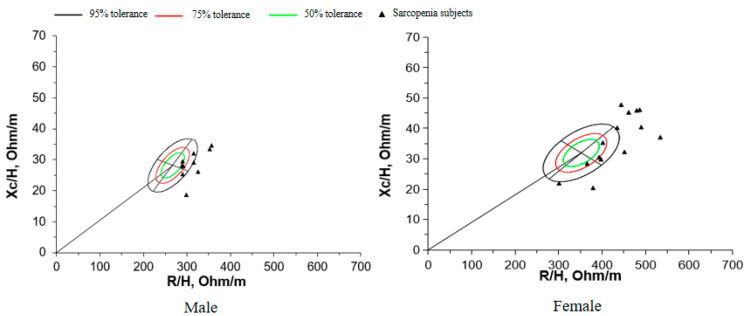
The specific vectors of individual sarcopenia subjects plotted on the bivariate tolerance ellipses.

**Figure 3 sensors-23-07090-f003:**
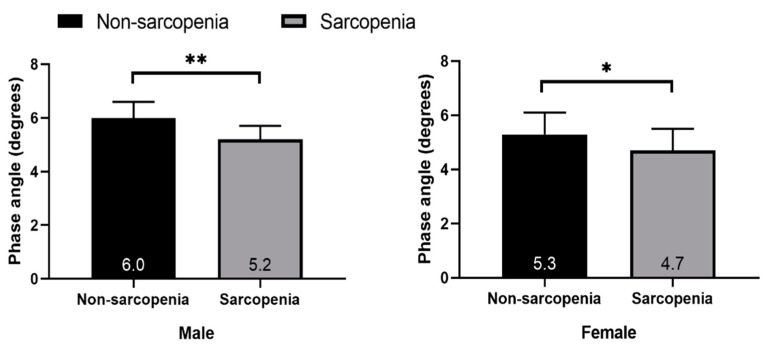
Phase angle in the sarcopenia and non-sarcopenia groups among males and females. * *p* < 0.05, ** *p* < 0.01, different from the group.

**Table 1 sensors-23-07090-t001:** Descriptive characteristics of the study population by sex.

	Males (n = 75)	Females (n = 71)	*p* Value
Age (years)	76.6 ± 4.2	75.1 ± 4.4	0.034
Height (cm)	166.5 ± 4.8	152.2 ± 5.1	<0.001
Weight (kg)	65.4 ± 7.2	53.2 ± 6.6	<0.001
BMI (kg/m^2^)	23.6 ± 2.3	22.9 ± 2.2	0.063
WB*_R* (Ω)	454.0 ± 42.4	557.4 ± 73.9	<0.001
WB_*Xc* (Ω)	47.1 ± 5.9	49.6 ± 8.3	0.028
WB_PhA (°)	5.9 ± 0.7	5.1 ± 0.6	<0.001
*R*/*H* (Ω/m)	272.9 ± 27.5	366.7 ± 51.1	<0.001
*Xc*/*H* (Ω/m)	28.3 ± 3.8	32.6 ± 5.3	<0.001
ASM (kg)	20.7 ± 2.3	14.0 ± 1.6	<0.001
ASM/H^2^ (kg/m^2^)	7.4 ± 0.7	6.0 ± 0.5	<0.001
HGS (kg)	31.6 ± 5.2	22.0 ± 3.0	<0.001
Physical performance (m/s)	1.0 ± 0.2	1.0 ± 0.2	0.421
Muscle quality (kg/kg)	1.64 ± 0.18	1.55 ± 0.17	0.006

*H*, height; BMI, body mass index; ASM, appendicular skeletal muscle; HGS, handgrip strength; WB, whole body; *R*, resistance; *Xc*, reactance; PhA, phase angle. Physical performance was assessed as the 6 m walk speed; muscle mass was calculated by dividing handgrip strength by appendicular skeletal muscle.

**Table 2 sensors-23-07090-t002:** Impedance and PhA components in sarcopenia and non-sarcopenia subjects by sex.

		n	*R*	*Xc*	PhA
M	Non-sarcopenia	65	444.9 ± 35.9	47.0 ± 5.4	6.0 ± 0.6
	Sarcopenia	10	513.7 ± 31.9	47.9 ± 8.9	5.2 ± 0.5 **
F	Non-sarcopenia	57	536.7 ± 44.7	48.5 ± 5.4	5.3 ± 0.8
	Sarcopenia	14	655.0 ± 93.4	54.7 ± 14.6	4.7 ± 0.8 *

M, male; F, female; *R*, resistance; *Xc*, reactance; PhA, phase angle. The differences between non-sarcopenia and sarcopenia groups were evaluated with the Mann–Whitney U test; * *p* < 0.05; ** *p* < 0.01. Sarcopenia was diagnosed according to the following criteria (AWGS): (1) low muscle strength (HGS, M < 28 kg, F < 18 kg) + (2) low ASM (DXA, M < 7.0 kg/m^2^, F < 5.4 kg/m^2^) or (3) low physical performance (6 m walk < 1.0 m/s).

**Table 3 sensors-23-07090-t003:** Impedance vector components in sarcopenia and non-sarcopenia subjects by sex.

		n	*R/H*	*Xc/H*	*r*	*H*	PhA
M	Non-sarcopenia	65	266.9 ± 22.4	28.2 ± 3.4	0.63	1.67 ± 0.05	6.0 ± 0.6
	Sarcopenia	10	312.1 ± 25.8	29.1 ± 5.8	0.71	1.65 ± 0.04	5.2 ± 0.5 **
F	Non-sarcopenia	57	353.0 ± 34.2	31.9 ± 3.7	0.54	1.52 ± 0.05	5.3 ± 0.8
	Sarcopenia	14	429.9 ± 61.0	35.8 ± 9.0	0.75	1.52 ± 0.04	4.7 ± 0.8 *

M, male; F, female; *H*, height; *R*, resistance; *Xc*, reactance; PhA, phase angle; r, correlation coefficient between *R*/*H* and *Xc*/*H*. The differences between non-sarcopenia and sarcopenia groups were evaluated with the Mann–Whitney U test; * *p* < 0.05; ** *p* < 0.01. Sarcopenia was diagnosed according to the following criteria (AWGS): (1) low muscle strength (HGS, M < 28 kg, F < 18 kg) + (2) low ASM (DXA, M < 7.0 kg/m^2^, F < 5.4 kg/m^2^) or (3) low physical performance (6 m walk < 1.0 m/s).

**Table 4 sensors-23-07090-t004:** Correlation of PhA with age and diagnostic components of sarcopenia in males and females.

Gender	Variables	Pearson’s r	*p* Value
Male	Age	−0.275 *	0.017
	ASM	0.229 *	0.048
	ASM/H^2^	0.352 **	0.002
	HGS	0.350 **	0.002
	Physical performance	0.493 ***	<0.001
	Muscle quality	0.272 *	0.018
Females	Age	−0.389 **	0.002
	ASM	0.271 *	0.038
	ASM/H^2^	0.235	0.073
	HGS	0.640 ***	<0.001
	Physical performance	0.074	0.578
	Muscle quality	0.524 ***	<0.001

ASM, appendicular skeletal muscle mass; *H*, height; HGS, handgrip strength. *: *p* < 0.05, ** *p* < 0.01, *** *p* < 0.001.

**Table 5 sensors-23-07090-t005:** Linear regression analysis between the phase angle and the dependent variables.

Dependent Variable		*R* ^2^	β	*p* Value	95% CI	Partial Correlation Coefficient
Male	Muscle strength	0.12	2.8	<0.001	1.060–4.579	0.35
	Muscle quality	0.07	0.07	0.037	0.013–0.139	0.27
	Physical performance	0.24	0.15	<0.001	0.088–0.211	0.49
Female	Muscle strength	0.41	3.31	<0.001	2.282–4.496	0.64
	Muscle quality	0.27	0.16	<0.001	0.090–0.227	0.52
	Physical performance	0.005	0.02	0.543	−0.062–0.109	0.07

*R*^2^, Coefficient of determination; β, unstandardized coefficient; CI, confidence interval.

**Table 6 sensors-23-07090-t006:** Multiple linear regression analysis on the phase angle.

	Model 1				Model 2				Model 3			
	95% CI				95% CI				95% CI			
Males	Lower	Upper	β	*p* Value	Lower	Upper	β	*p* Value	Lower	Upper	β	*p* Value
Physical performance	0.953	2.289	1.62	<0.001	0.767	2.094	1.43	<0.001	0.693	1.996	1.34	<0.001
ASM/H^2^	-	-	-	-	0.050	0.455	0.25	0.015	0.053	0.448	0.25	0.01
Muscle quality	-	-	-	-	-	-	-	-	0.069	1.452	0.76	0.03
Females												
Muscle quality	0.985	2.481	1.73	<0.001	1.375	2.741	2.06	<0.001	-	-	-	-
ASM	-	-	-	-	0.078	0.231	0.16	<0.001	-	-	-	-

ASM, appendicular skeletal muscle; H, height. In males, Model 1 had *R*^2^ = 0.243, Model 2 had *R*^2^ = 0.303, and Model 3 had *R*^2^ = 0.347. In females, Model 1 had *R*^2^ = 0.274 and Model 2 had *R*^2^ = 0.439.

## Data Availability

The datasets generated and/or analyzed during the study will be available from the corresponding author upon reasonable request.

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
