# Peer review of "Distribution of Bioelectrical Impedance Vector Analysis and Phase Angle in Korean Elderly and Sarcopenia"

_sensors, 2023, doi:10.3390/s23167090_

Round 1

Reviewer 1 Report

The Introduction, although long, provides an excellent background to the BIVA method and why the present study was undertaken.

Material and method:

This is a very interesting manuscript with useful data however it's very important to report the methodology.

2.3. Measurement of the bioelectrical impedance parameters

it's necessary to describe who perform the subject's evaluation and if there were a previous standardization ( when in the morning , how many minutes in supine position, etc )   ref.

Please include the characteristics of electrodes used for BIA evaluation.
The accuracy of the bia analyzers cannot be validated but can be verified. Have you made some test retest or CV% assessment  ?

 Discussion

 We included  this study in the discussion

“ A study of 1,009 Japanese community-dwelling elderly individuals demonstrated that PhA was moderately correlated with muscle strength and walking speed [45]. “

 I suggest to remove any studies where pha is measured with different technology and method. Remember that pha is a instrument specific measurement.

Bia 101 rjl  showed to measure significantly different the phase angle compared to Tanita MC780 , with different sensitivity to changes  .

Vermeulen-Serpa, Karina Marques, et al. "Gender-specific bioelectrical impedance reference values in healthy children." Clinical Nutrition Open Science 49 (2023): 77-87.

Martins, Priscila Custódio, et al. "Phase Angle And Body Composition: A Scoping Review." Clinical Nutrition ESPEN (2023).

In limitation is necessary to mention that such data is useful only in Korean population.

Please have a look to these papers.

Lukaski, Henry C., et al. "Normal Reference Plots of the Bioelectrical Impedance Vector for Healthy Korean Adults." Journal of Korean Medical Science 34.40 (2019).

Ethnicity has been shown to influence BIVA  displacement or distribution - see DOI: 10.1111/j.1749-6632.2000.tb06449

 Another very important limitation is that Phase angle is a marker that suffer different level of tissue hydration,  so unless you use this parameter without BIVA information ( vector length)  the phase angle lose specificity to sarcopenia detection.

Bellido, Diego, et al. "Future lines of research on phase angle: strengths and limitations." Reviews in Endocrine and Metabolic Disorders (2023): 1-21.

Mistyping :

Below the fig.3  the paragraph indicates the abbreviations :

“(2) vectors lying on the left side and above the major axis or on the right side and below the major axis of tolerance ellipses indicate more or less soft tissue mass, respectively. : Resistance; Xc: reactance; :height[46]”

 You misses including R, Xc ; H

Author Response

REVIEWERS' COMMENTS: (Reviewer 1)

The Introduction, although long, provides an excellent background to the BIVA method and why the present study was undertaken.

SPECIAL COMMENTS

Comment

Response

Revision

1.       It's necessary to describe who perform the subject's evaluation and if there were a previous standardization ( when in the morning , how many minutes in supine position, etc )   ref.

Thanks for your comment. We added this section.

(The last meal was completed at least 4 hours prior to the measurement and the bladder was emptied within 30 minutes prior to the measurement. Bioelectric variables were measured by skilled operators according to international standards. [48]. Each evaluation was performed by the same tester. Intraclass correlation coefficient (ICC) of the examiner for the repeatability was conducted for the single test. The intraclass correlation coefficients of the examiner for the repeatability were 0.961 and 0.955, respectively as reported [28].)

Before: No

After: See the line 117-122.

2.       We included  this study in the discussion “ A study of 1,009 Japanese community-dwelling elderly individuals demonstrated that PhA was moderately correlated with muscle strength and walking speed [45]. ” I suggest to remove any studies where pha is measured with different technology and method. Remember that pha is a instrument specific measurement.

Thanks for your comment.We have removed this section.

Before: No

After: Deleted

3.       In limitation is necessary to mention that such data is useful only in Korean population.

Thanks for your comment. We have added this limitation.

(Finally, these reference data are only applicable to specific devices and specific populations. Caution should be exercised when applying to other populations and different devices.)

Before: No

After: see the line 306-307.

4.       Mistyping :Below the fig.3  the paragraph indicates the abbreviations :“(2) vectors lying on the left side and above the major axis or on the right side and below the major axis of tolerance ellipses indicate more or less soft tissue mass, respectively. : Resistance; Xc: reactance; :height[46]” You misses including R, Xc ; H.

Thanks for your comment. We corrected these sections.

Before: Mistyping.

After: See the line 218-219.

Reviewer 2 Report

The manuscript has two aims: to verify the ability of BIVA provides useful data to evaluate sarcopenia in elderly; and to evaluate the relationships between PhA and physical body properties. Significant number of volunteering subjects have been recruited, sarcopenia diagnosed based on the EWGSOP2 guidelines and BIA performed together with physical parameter detection. Variety of statistical analysis methods are applied to the acquired data to test the hypotheses of the usability of BIVA and PhA in sarcopenia characteristics.

Reviewer has several comments and concerns:

1. Major issue is that the line numbers are missing from the manuscript - this is quite hard for the reviewer to review and refer to certain lines. Please add the line numbers, this is crucial.

2. The abstract of the paper is in unacceptable format. First of all, please use the format, that is defined in the Author Instructions: 1) Background: 2) Methods: 3) Results: 4) Conclusion. Secondly, the abstract should be a total of about 200 words maximum. This must be reworked and reduced more than 200 words. Thirdly, there are abbreviation in the Abstract, that are not defined. In Abstract, you also need to fully write the words once, and then define the abbreviation.

3. The reviewer is worried about the novelty of the manuscript. A sentence from the conclusions: "The results show that BIVA can detect changes in muscle mass in individuals with sarcopenia and is a practical method for assessing sarcopenia in the field." The same conclusion is drawing also for example in this paper, published more than 10 years ago: Marini E, Buffa R, Saragat B, et al. The potential of classic and specific bioelectrical impedance vector analysis for the assessment of sarcopenia and sarcopenic obesity. Clin Interv Aging. 2012;7:585-591. doi:10.2147/CIA.S38488. Please state clearly the novelty in the manuscript!

4. Sentence in the manuscript: "Adhesive-gel electrodes were placed at defined anatomical sites on the anterior of the dorsal surfaces of the right hand, wrist, ankle, and foot." Reviewer comment: please define these defined anatomical sites too. Where and how were these adhesive-gel electrodes attached? I suggest to add an illustration as as the written section in page 4 is quite hard to follow?

5. Sentence: "Males were significantly taller and heavier than females (p<0.001), and females had significantly higher R, R/H, and Xc/H than males (p<0.001)" in page 4 and sentence "Males had significantly greater height, weight, ASM, ASMI, muscle quality, and PhA than females (p<0.001), and females had significantly higher R than males (p<0.001)." in page 7 say the same thing. Is this repetition necessary?

6. Figures 1-3 are of very low quality, the texts and numbers are sporadically unclear. Please provide figures of acceptable quality.

7. In several places in the manuscript, the terms "sarcopenia group" and "non-sarcopenia" groups are stated. However, no definition or description of the designation into each group is described in the manuscript. First sentence about non-sarcopenia group appears in page 5 just after Table 1 without any pre-knowledge. Please define.

8. Table 2 shows, that the total number of female sarcopenia patients is 2? Is it really so? Based on that, no any reliable statistical analysis can be performed. Please explain.

9. Table 1 and 4 are duplicating each other in several lines. Even more, some numbers are different, unless, should be duplicates. Why is this duplication necessary, explanation of revision required.

10. Table 2 and 5 are duplicating each other. What is the reason to duplicate the results in the single manuscript?

11. Figure 5 is not referenced in the text. All the figures/tables must be referenced in the text.

12. Sentence in the beginning of Discussion section: "This study used the AWGS criteria to diagnose sarcopenia in Korean elderly individuals, established BIVA reference values in 122 healthy Korean elderly people aged 69-91 years, and determined whether BIVA can be used to diagnose sarcopenia in individuals." Where does this number 122 elderly people comes from? Previously you have stated the participation of 134 elderly volunteers...

Some very long sentences in the manuscript but generally the English is OK.

Author Response

REVIEWERS' COMMENTS: (Reviewer 2)

The manuscript has two aims: to verify the ability of BIVA provides useful data to evaluate sarcopenia in elderly; and to evaluate the relationships between PhA and physical body properties. Significant number of volunteering subjects have been recruited, sarcopenia diagnosed based on the EWGSOP2 guidelines and BIA performed together with physical parameter detection. Variety of statistical analysis methods are applied to the acquired data to test the hypotheses of the usability of BIVA and PhA in sarcopenia characteristics.

SPECIAL COMMENTS

Comment

Response

Revision

1. Major issue is that the line numbers are missing from the manuscript - this is quite hard for the reviewer to review and refer to certain lines. Please add the line numbers, this is crucial.

Thanks for your comment. e added the line numbers.

Before: No

After: See the manuscript.

2. The abstract of the paper is in unacceptable format. First of all, please use the format, that is defined in the Author Instructions: 1) Background: 2) Methods: 3) Results: 4) Conclusion. Secondly, the abstract should be a total of about 200 words maximum. This must be reworked and reduced more than 200 words. Thirdly, there are abbreviation in the Abstract, that are not defined. In Abstract, you also need to fully write the words once, and then define the abbreviation.

Thanks for your comment. We adjusted the abstract section and defined the abbreviations.

Before: unacceptable format.

After: See the Abstract.

3. The reviewer is worried about the novelty of the manuscript. A sentence from the conclusions: "The results show that BIVA can detect changes in muscle mass in individuals with sarcopenia and is a practical method for assessing sarcopenia in the field." The same conclusion is drawing also for example in this paper, published more than 10 years ago: Marini E, Buffa R, Saragat B, et al. The potential of classic and specific bioelectrical impedance vector analysis for the assessment of sarcopenia and sarcopenic obesity. Clin Interv Aging. 2012;7:585-591. doi:10.2147/CIA.S38488. Please state clearly the novelty in the manuscript!

Thanks for your comment. Our study is necessary because different populations and different BIA manufacturing companies can lead to variability in the study results.Currently, simulated impedance data for Korean samples are generated in multi-spectrum devices, and 50khz phase-sensitive devices generate reference values for Korean elderly people that have not yet been shown in studies.

(Ref: Lukaski HC. Letter to the Editor: Normal Reference Plots of the Bioelectrical Impedance Vector for Healthy Korean Adults. J Korean Med Sci. 2019;34(40):e274. Published 2019 Oct 21. doi:10.3346/jkms.2019.34.e274)

Before: The results show that BIVA can detect changes in muscle mass in individuals with sarcopenia and is a practical method for assessing sarcopenia in the field.

After: See the line 314-316.

“This study established BIVA reference data for Korean elderly using 50khz phase-sensitive equipment, in addition to detecting changes in muscle mass in Korean elderly patients with sarcopenia, providing evidence that BIVA can be used as a field assessment method for Korean elderly with sarcopenia.”

 4. Sentence in the manuscript: "Adhesive-gel electrodes were placed at defined anatomical sites on the anterior of the dorsal surfaces of the right hand, wrist, ankle, and foot." Reviewer comment: please define these defined anatomical sites too. Where and how were these adhesive-gel electrodes attached? I suggest to add an illustration as as the written section in page 4 is quite hard to follow?

Thanks for your comment. For details we have described in a previously published study, please refer to the article:

Jiang F, Tang S, Eom J-J, Song K-H, Kim H, Chung S, Kim C-H. Accuracy of Estimated Bioimpedance Parameters with Octapolar Segmental Bioimpedance Analysis. Sensors. 2022; 22(7):2681. https://doi.org/10.3390/s22072681

5. Sentence: "Males were significantly taller and heavier than females (p<0.001), and females had significantly higher R, R/H, and Xc/H than males (p<0.001)" in page 4 and sentence "Males had significantly greater height, weight, ASM, ASMI, muscle quality, and PhA than females (p<0.001), and females had significantly higher R than males (p<0.001)." in page 7 say the same thing. Is this repetition necessary?

Thanks for your comment. We removed the duplicate descriptions.

Before: The duplicate description.

After: See the results section.

6. Figures 1-3 are of very low quality, the texts and numbers are sporadically unclear. Please provide figures of acceptable quality.

Thanks for your comment. We replaced the image with a clearer one.

Before: Figures 1-3 are of very low quality.

After: See the Figures 1-3.

7. In several places in the manuscript, the terms "sarcopenia group" and "non-sarcopenia" groups are stated. However, no definition or description of the designation into each group is described in the manuscript. First sentence about non-sarcopenia group appears in page 5 just after Table 1 without any pre-knowledge. Please define.

Thanks for your comment. We have added this section.

Before: No

After: See the line 133-134

8. Table 2 shows, that the total number of female sarcopenia patients is 2? Is it really so? Based on that, no any reliable statistical analysis can be performed. Please explain.

Thanks for your comment. Since the sample size of patients with sarcopenia was less than 10, we used a non-parametric test.

9. Table 1 and 4 are duplicating each other in several lines. Even more, some numbers are different, unless, should be duplicates. Why is this duplication necessary, explanation of revision required.

Thanks for your comment. We combined Tables 1 and 4.

Before: Table 1 and 4.

After: See the Table 1.

10. Table 2 and 5 are duplicating each other. What is the reason to duplicate the results in the single manuscript?

Thanks for your comment. We removed the Table 5. See the manuscript.

Before: Table 2 and 5.

After: Removed the Table5.

11. Figure 5 is not referenced in the text. All the figures/tables must be referenced in the text.

Thanks for your comment. We cited Figure 5 in the text.

Before: No

After: See the line 248.

12. Sentence in the beginning of Discussion section: "This study used the AWGS criteria to diagnose sarcopenia in Korean elderly individuals, established BIVA reference values in 122 healthy Korean elderly people aged 69-91 years, and determined whether BIVA can be used to diagnose sarcopenia in individuals." Where does this number 122 elderly people comes from? Previously you have stated the participation of 134 elderly volunteers...

Because 12 elderly patients with sarcopenia were excluded, we established reference values for BIVA using 122 healthy Korean elderly people aged 69-91 years.

Round 2

Reviewer 1 Report

thank you for all the effort to improve the previous version . I have no issues 

Author Response

This study has bee improved by your comments.

Thanks you for your admission.

Reviewer 2 Report

Reviewer thanks the authors for the next version of manuscript. Unless the Abstract is now in an acceptable length and structure, my concerns are the same as in first round.

My concerns are listed followingly:

1. (point 1 in first review round) Unless the authors claim that the line numbers are added to the manuscript documents, in the submitted file, there are no line numbers.

2. (point 3 in first review round) My concern concerning the novelty of the manuscript remain, as the explanation (and respective amendment of the manuscript) of the authors has not been convincing. I understand that the differences in measurement of BIVA and PhA data may be related to ethnic groups. However, if the target is the comparison between the BIVA and PhA data of different ethnic groups, the comparison is missing (and clear indication in the title and text). The claim in the response file from the authors that currently the impedance data is generated by using a multi-spectrum devices and for the manuscript, 50 kHz device is used, is not something to rely on. Multi-spectrum in essence means variety of excitation frequencies (spectrum), not only a single on (like 50 kHz). If you compare the data of difference devices, it must be clearly presented as a comparison. My question, what is the new knowledge, that you gain with single frequency of the same data is available also in multi-spectrum device? Also, novelty, or the demand (what you aim to achieve with the research and the planned innovative outcomes must be named already in the introduction section. Not only in the conclusions, without no previous insight etc. I claim, that the revised manuscript still does not have enough novelty, as the last sentence in the Introductory section has not been revised by the authors. Such body of knowledge, as the title of the manuscript declares is answered in previous works in scientific literature:

Kołodziej, M.; Kozieł, S.; Ignasiak, Z. The Use of the Bioelectrical Impedance Phase Angle to Assess the Risk of Sarcopenia in People Aged 50 and above in Poland. Int. J. Environ. Res. Public Health 202219, 4687. https://doi.org/10.3390/ijerph19084687

Elisabetta Marini, Roberto Buffa, Bruno Saragat, Alessandra Coin, Elena Debora Toffanello, Linda Berton, Enzo Manzato & Giuseppe Sergi (2012) The potential of classic and specific bioelectrical impedance vector analysis for the assessment of sarcopenia and sarcopenic obesity, Clinical Interventions in Aging, 7:, 585-591, DOI: 10.2147/CIA.S38488

Marini E, Buffa R, Saragat B, et al. The potential of classic and specific bioelectrical impedance vector analysis for the assessment of sarcopenia and sarcopenic obesity. Clin Interv Aging. 2012;7:585-591. doi:10.2147/CIA.S38488.

3. (point 7 in first review round) The author's reply concerning the division of participants into sarcopenia and non-sarcopenia groups is not satisfactory. What was the criteria to divide participants into these groups? This must be clearly stated in the manuscript, not only in the answer to the reviewer.

4. (point 8 in first review round) My comment about female sarcopenia patients in Table 2. Ok, you have used non-parametric test. This has to be outlined in the manuscript too, as the statistical results without any such comment are misleading. Such results (based on 2 samples) are not comparable with the cases where the sample size for statistical analysis is significant.

5. (point 12 in first review round) The author's answer concerning the number of participants, recruited into the research is not directed properly. Figure 1 in the revised manuscript declares clearly that n=134 participants were included in the analysis. While the first sentence in the Discussion section says that 122 healthy Korean elderly people were involved. No information in the text that 12 were additionally excluded.

Some typography and grammar mistakes.

Author Response

Comments and Suggestions for Authors: Reviewer thanks the authors for the next version of manuscript. Unless the Abstract is now in an acceptable length and structure, my concerns are the same as in first round.

My concerns are listed followingly:

  1. (point 1 in first review round) Unless the authors claim that the line numbers are added to the manuscript documents, in the submitted file, there are no line numbers.

response 1: The line number are in the manuscript because we wrote the paper from a sonders’ template. Please check again.

  1. (point 3 in first review round) My concern concerning the novelty of the manuscript remain, as the explanation (and respective amendment of the manuscript) of the authors has not been convincing.I understand that the differences in measurement of BIVA and PhA data may be related to ethnic groups. However, if the target is the comparison between the BIVA and PhA data of different ethnic groups, the comparison is missing (and clear indication in the title and text).

response 2-1: This study is not intended to compare differences between ethnic groups.

The purpose of this study was to confirm the distribution of phase angle and BIVA in elderly Koreans and to investigate whether there was a difference in phase angle and BIVA between normal elderly and sarcopenic patients in Korean elderly. However, your comment of the title is convincing, so we have corrected the title as "Distribution of bioelectrical impedance vector analysis and phase angle in Korean elderly and sarcopenia."

 The claim in the response file from the authors that currently the impedance data is generated by using a multi-spectrum devices and for the manuscript, 50 kHz device is used, is not something to rely on. Multi-spectrum in essence means variety of excitation frequencies (spectrum), not only a single on (like 50 kHz). If you compare the data of difference devices, it must be clearly presented as a comparison. My question, what is the new knowledge, that you gain with single frequency of the same data is available also in multi-spectrum device? 

response 2-2: In this study, we do not intend to convert meanings obtained from single frequencies to multi-frequency ones. However, since the phase angle obtained at multiple frequencies is most amplified for 50 kHz, this frequency is only used in the sensitive range. Like what has been reported in studies so far, only 50 kHz of multiple frequencies is used. Also, in the two papers you presented (the second and third papers are the same paper), the phase angle was studied at a single frequency of 50 kHz.

 I claim that the revised manuscript still does not have enough novelty, as the last sentence in the Introductory section has not been revised by the authors. Such body of knowledge, as the title of the manuscript declares is answered in previous works in scientific literature:

response 2-3:  Thank you for your comment. We have rewritten the last sentence and title as following. (Title) Distribution of bioelectrical impedance vector analysis and phase angle in Korean elderly and sarcopenia. (the last sentence in introduction) This study aims to identify the distribution of BIVA and comparison of phase angle between Korean elderly and sarcopenia for trying to determine whether BIVA and phase angle could be a useful biomarker in identifying sarcopenia in Korean population.

Kołodziej, M.; Kozieł, S.; Ignasiak, Z. The Use of the Bioelectrical Impedance Phase Angle to Assess the Risk of Sarcopenia in People Aged 50 and above in Poland. Int. J. Environ. Res. Public Health 2022, 19, 4687. https://doi.org/10.3390/ijerph19084687

Elisabetta Marini, Roberto Buffa, Bruno Saragat, Alessandra Coin, Elena Debora Toffanello, Linda Berton, Enzo Manzato & Giuseppe Sergi (2012) The potential of classic and specific bioelectrical impedance vector analysis for the assessment of sarcopenia and sarcopenic obesity, Clinical Interventions in Aging, 7:, 585-591, DOI: 10.2147/CIA.S38488

Marini E, Buffa R, Saragat B, et al. The potential of classic and specific bioelectrical impedance vector analysis for the assessment of sarcopenia and sarcopenic obesity. Clin Interv Aging. 2012;7:585-591. doi:10.2147/CIA.S38488.

Response 2-4: Thank you for your comment. We reviewed the three articles above and confirmed the phase angle have been used only one frequency as 50kHz. (The second and third article is the same.)

  1. (point 7 in first review round) The author's reply concerning the division of participants into sarcopenia and non-sarcopenia groups is not satisfactory. What was the criteria to divide participants into these groups? This must be clearly stated in the manuscript, not only in the answer to the reviewer.

Response 3: We have described the subjects included in this study and the process of dividing into those with and without sarcopenia according to the EWGSOP2 guidelines was described. Please check lines 106 – 110 and 135-146 in the manuscript.

(Lines 106 – 110) The inclusion criteria were aged 65 or more, as well as independence and autonomy in everyday life. Participants were evaluated to be subjectively healthy based on declarations of good health and no limitations in daily activities. Of the 220 people registered for the test, 74 were exclude. Exclusion criteria were as follows: those who did not perform any tests, limb amputation, the presence of metal prosthetic devices, electronically based implants.

(Lines 135 – 146) Sarcopenia was diagnosed as low muscle strength and low muscle mass and/or low physical performance according to the EWGSOP2 guidelines [9]. Appendicular skeletal muscle (ASM) was assessed by DXA. All centers used the same DXA model (Lunar DPX-L model, software version 3.4) for body composition measurements. The measurements were performed in medium scan mode with the subject lying in a supine position. The scanning time was approximately 20 min. Low muscle mass was defined by DXA < 7.0 kg/m2 in males and <5.9 kg/m2 in females for the Korean population [31]. Grip strength was tested with a grip dynamometer. Muscle strength was assessed through handgrip testing with a Grip-D hand-held grip dynamometer (Takei, Niigata, Japan). The strength was recorded in kilograms (kg). The left and right hands were tested 3 times, and the highest values were used [29]. Low muscle strength was defined as handgrip strength <28 kg for males and <18 kg for females for the Korean population. Physical performance was assessed by asking participants to walk 6 meters at a comfortable pace. The test started when their first foot crossed the starting line and stopped when their second foot crossed the finish line [30]. The 6-meter walk was performed 2 times, and the highest value was used. The criteria for low physical performance were 6-meter walk speed <1.0 m/s [30].

  1. (point 8 in first review round) My comment about female sarcopenia patients in Table 2. Ok, you have used non-parametric test. This has to be outlined in the manuscript too, as the statistical results without any such comment are misleading. Such results (based on 2 samples) are not comparable with the cases where the sample size for statistical analysis is significant.

Response 4: We have included data from additional studies. An explanation of additional research data and the number of IRB were written in the Material and Methods. We have reanalyzed and rewritten the results. Thanks for pointing out to write a good research paper.

  1. (point 12 in first review round) The author's answer concerning the number of participants, recruited into the research is not directed properly. Figure 1 in the revised manuscript declares clearly that n=134 participants were included in the analysis. While the first sentence in the Discussion section says that 122 healthy Korean elderly people were involved. No information in the text that 12 were additionally excluded.

Response 5: in order to clearly revise the point you pointed out, Figure 1 and its explanations were erased from the results and described again in detail in the section of “Materials and Methods”. Please check lines 106 – 110.

(Lines 104 – 110) The study was approved by the Ethics Committee of the Korean National Sport University (No. 1263-201903-HR-010-02) and Soonchunhyung University (1040875-2020307-BC-083) and performed in accordance with the Declaration of Helsinki.

The inclusion criteria were aged 65 or more, as well as independence and autonomy in everyday life. Participants were evaluated to be subjectively healthy based on declarations of good health and no limitations in daily activities. Of the 220 people registered for the test, 74 were exclude. Exclusion criteria were as follows: those who did not perform any tests, limb amputation, the presence of metal prosthetic devices, or electronically based implants.

Round 3

Reviewer 2 Report

Reviewer thanks the authors for the answers and the respective amendments in the manuscript. I am mostly satisfied, but there are still two point that needs addressing:

1. There is still confusion with the number of involved participant of the study. Section 2.1 Participants declare: from 220 people, 74 were excluded. So, based on that, 146 people were included. The same number can be derived from Table 1. But at the same time, the section 4. Discussion claims: 122 healthy Korean elderly people. I assume this is either a typo (1) or there is some information missing regarding the real number of participants (2)!

2. The novelty of the manuscript is still sparsely understandable from the manuscript itself. The authors have explained this in the answers to the reviewer as: "50khz phase-sensitive devices generate reference values for Korean elderly people that have not yet been shown in studies". But this needs a clear declaration in the manuscript as well. By using statements, the novelty of the current paper is..., the innovation is...

English language of the manuscript contains some typographic mistakes.

Author Response

Reviewer thanks the authors for the answers and the respective amendments in the manuscript. I am mostly satisfied, but there are still two point that needs addressing:

  1. There is still confusion with the number of involved participant of the study. Section 2.1 Participants declare: from 220 people, 74 were excluded. So, based on that, 146 people were included. The same number can be derived from Table 1. But at the same time, the section 4. Discussion claims: 122 healthy Korean elderly people. I assume this is either a typo (1) or there is some information missing regarding the real number of participants (2)!

 Resonse) Thank you for your comment. As you comment, it was mistyped. It has been corrected. Thank you. See line number 1017. 

2. The novelty of the manuscript is still sparsely understandable from the manuscript itself. The authors have explained this in the answers to the reviewer as: "50khz phase-sensitive devices generate reference values for Korean elderly people that have not yet been shown in studies". But this needs a clear declaration in the manuscript as well. By using statements, the novelty of the current paper is..., the innovation is...

Response) We think your comment is valuable. We have added the innovative meanings in the introduction of this study as followings:

(Line 99-101) Bioelectric variables at 50kHz include resistance and induction resistance, the relationships between which indicate the physiological state of the human body. The variables directly measured by 50kHz phase-sensitive BIA are resistance and sensory reactance.  

(Line 384-382) These results provide new evidence for the validity of 50kHz phase-sensitive bioelectrical measurements as biomarkers of hydration status, catabolic states, and nutritional status. To enable the development of clinical research and practice in sarcopenia, it is essential to establish distribution of BIVA and phase angle for a population and disease status. However, validation studies that use BIVA and phase angle to identify individuals with sarcopenia in the Korean population are currently lacking.

 This study aimed to identify the distribution of BIVA and comparison of the phase angle between Korean elderly and patients with sarcopenia in an attempt to identifying whether BIVA and phase angle at 50kHz could be an innovative useful biomarkers in identifying sarcopenia in the Korean population.

Comments on the Quality of English Language
English language of the manuscript contains some typographic mistakes.

response) We reviewed and revised this manuscript by Editage, a professional English editing company (www.editage.co.kr). The revised contents can be found in the manuscript.